# Genomic and Socioeconomic Determinants of Racial Disparities in Breast Cancer Survival: Insights from the All of Us Program

**DOI:** 10.3390/cancers16193294

**Published:** 2024-09-27

**Authors:** Nubaira Rizvi, Hui Lyu, Leah Vaidya, Xiao-Cheng Wu, Lucio Miele, Qingzhao Yu

**Affiliations:** 1Biostatistics and Data Science, School of Public Health, LSU Health—New Orleans, New Orleans, LA 70112, USA; nrizvi@lsuhsc.edu (N.R.); lvaidy@lsuhsc.edu (L.V.); 2Department of Interdisciplinary Oncology, LSU Health—New Orleans, New Orleans, LA 70112, USA; hlyu@lsuhsc.edu; 3Louisiana Tumor Registry, School of Public Health, LSU Health—New Orleans, New Orleans, LA 70112, USA; xwu@lsuhsc.edu; 4Department of Genetics, School of Medicine, LSU Health—New Orleans, New Orleans, LA 70112, USA; lmiele@lsuhsc.edu

**Keywords:** breast cancer, racial disparity, genomic mutation, mediation analysis, All of US

## Abstract

**Simple Summary:**

Research shows that Black women generally have poorer breast cancer survival rates than White women in the U.S. Our study aims to identify genomic variants and socioeconomic determinants that might explain this racial disparity using mediation analysis. Based on the All of US research program, we identified 15 gene mutations along with factors age, general health, and general quality of life that can explain the observed racial disparity. By studying how these genes behave differently in Black and White breast cancer patients, researchers can gain important insights into the mechanism underlying the cancer development and prognosis among different populations. This understanding could help create better, personalized treatments, especially to address the differences in breast cancer outcomes among racial groups.

**Abstract:**

**Background**: Breast cancer outcomes are worse among Black women in the U.S. compared to White women. While extensive research has focused on risk factors contributing to breast cancer; the role of genomic elements in health disparities between these racial groups remains unclear. This study aims to identify genomic variants and socioeconomic status (SES) determinants influencing racial disparities in breast cancer survival through multiple mediation analyses. **Methods**: Our investigation is based on the NIH-supported All of Us (AoU) program and analyzes 7452 female participants with malignant tumors of breast, including 5073 with genomic data. A log-rank test reveals significant racial differences in overall survival time between Black and White participants (*p*-value = 0.04). Multiple mediation analysis examines the effects of 9481 genetic variables across 23 chromosomes in explaining the racial disparity in survival, adjusting for SES variables. **Results**: 15 gene mutations, in addition to age, general health, and general quality of life, have significant effects (*p*-values < 0.001) in explaining the observed racial disparity. Mutations in TMEM132B, NARFL, SALL1, PAD12, RIPK1, ASB14, DCX, GNB1L, ARHGAP32, AL135787.1, WBP11, SLC16A12AS1, AP000345.1, IKBKB, and SUPT20H have significantly different distributions between Black and White participants. The disparity is completely explained by the included variables as the direct effect is insignificant (*p*-value = 0.73). **Conclusions**: The combined impact of SES determinants and genetic mutations can explain the observed differences in breast cancer survival among Black and White participants. Future studies will explore pathways and design in vivo and in vitro experiments to validate the functions of these genes

## 1. Introduction

Breast cancer is the most common cancer among women in the United States (US) [1]. If not detected and treated early, the cancerous cells can spread to other parts of the body, leading to more serious health complications. Risk factors for developing breast cancer include genetics, hormonal influences, lifestyle choices, and environmental factors [2]. Racial disparities in breast cancer outcomes are particularly concerning in the US. Studies have consistently shown Black women experience higher mortality rates from breast cancer despite having lower overall incidence rates than White women [3]. They are often diagnosed at more advanced stages when treatment options are more limited and less effective [4]. Moreover, they are more likely to be diagnosed with triple-negative breast cancer, a subtype that is particularly aggressive [5]. Disparities in socioeconomic status can exacerbate the issue due to disadvantaged living conditions and barriers to accessing timely and quality care among marginalized communities [6].

Researchers and policymakers are working toward a more comprehensive understanding of the underlying factors contributing to the observed disparities, with the goal of developing effective strategies to bridge this gap. Several genetic and non-genetic risk factors have been identified as contributors to the racial disparity in breast cancer survival. The genetic risk factors include mutations in BRCA, PR53, and PALB2 [7]. Non-genetic risk factors include socioeconomic status, access to healthcare, cultural and language barriers, geographic disparities, environmental factors, health behaviors and lifestyle choices, psychosocial factors, and biological differences [8].

Despite significant progress, gaps remain in our understanding of the risk factors. For example, many genetic variants that may be more prevalent in certain racial groups are understudied. Additionally, incomplete, or inconsistent data collection on race, ethnicity, and genetic information can hinder efforts to analyze the specific impact of genetic and non-genetic factors on breast cancer outcomes across different racial and ethnic groups. Understanding how these factors interact and contribute to disparities is a challenging task. Large-scale databases and comprehensive statistical methods are needed to differentiate the effects of individual or group risk factors.

In 2015, the National Institute of Health (NIH) initiated the All of Us (AoU) program as the largest precision medicine project ever undertaken in the US [9]. In this study, we take advantage of the comprehensive AoU dataset. Our research objective is to identify and differentiate genetic and socioeconomic risk factors that contribute to the observed racial disparity in overall survival among participants with breast cancer. We perform multiple mediation analyses to entangle the effects from different risk factors [10].

## 2. Materials and Methods

### 2.1. All-of-Us Program

The AoU project aims to accelerate health and medical advancements by inviting over a million participants from diverse backgrounds to build a rich data source [9]. The program currently offers access to almost 100,000 whole-genome sequences (WGS) that are highly diverse in populations [11]. Approximately 50% of the data originate from individuals belonging to racial or ethnic groups that have historically been less represented in research studies [12]. Approved researchers can use the secure cloud-based platform to access individual-level data, which includes genomic information within the controlled tier dataset.

From the AoU database, our cohort included 7452 participants who had malignant tumors of breast (ICD-10-CM diagnosis code C50) and were assigned female at birth. Among them, 1168 (15.67%) were Black and 6284 (84.33%) were White participants. All were included in the non-genetic analysis. Genomic data were available for 5073 of them, including 702 Black (13.84%) and 4371 White (86.16%) participants. All 5073 cases were included in this study jointly considering both genetic and socioeconomic risk factors.

#### 2.1.1. Socioeconomic Data

Table 1 lists the non-genetic variable names included in this study, and their descriptions, and values. It also lists the domain of the variables from the AoU database.

#### 2.1.2. Genomic Data

The genomic variant data for the breast cancer participants are accessed from the AoU Curated Data Repository (CDR) specifically from the C2022Q4R9 release, also known as v7 [13]. The AoU Data and Research Center provides Hail Matrix Tables (MT) with variants from different genome regions. We use short-read WGS data in MT that contain information on variants, such as Single Nucleotide Polymorphisms (SNPs), insertion, and deletion recorded in a compact data storage format [14,15]. This system consists of two files. One file has a matrix table where the rows represent variants, and the columns are subjects. The other, called “entries”, contains various fields, including “GT”, the genotype at each locus for every combination of variant and subject [14,16]. Samples are filtered to include those with phenotype value, race.

#### 2.1.3. Genetic Data Pre-Processing: Quality Control

Quality control (QC) steps for genomic data are crucial to ensure data accuracy, reliability, and consistency. The AoU program employs various QC measures such as initial data quality checks, variant calling QC, and low-quality data removal [17]. Figure 1 shows further steps performed to guarantee the data’s precision and dependability before analysis.

SNPs with more than 5% missing data were removed [18,19]. Sex concordance is already checked in the upstream genomic data QC process [16]. SNPs with a minor allele frequency (MAF) greater than 0.05 are retained. Deviations from Hardy–Weinberg equilibrium (HWE) indicate genotyping errors [18,19]. Variants with *p*-values from HWE tests greater than 10^−6^ are retained. Having a high or low heterozygosity rate can indicate inbreeding [19]. Individuals deviating more than 3 standard deviations from the mean heterozygosity rate are removed. Relatedness checks if a group of subjects is related to each other [19]. Relatedness flagged samples are removed. Additionally, the VCF filters field indicates the presence of variant filtering annotations generated by variant calling software. Variants with no filters applied are retained.

The QC steps help to mitigate potential sources of bias, genotyping errors, and confounding factors. After QC, the data are annotated using the variant annotation table (VAT), which includes the gene symbol annotations [16]. For each chromosome, the number of variants for each gene is calculated by counting the number of alternate alleles in the genotype (GT) field [20]. AoU workbench provides example codes for all the QC analyses. Following the QC, 9481 genes were retained across the 23 chromosomes for the 5073 selected subjects. Appendix A shows the number of genes for each chromosome.

### 2.2. Statistical Methods

When analyzing the relationship between a predictor (X, e.g., race) and a response variable (Y, e.g., overall survival/hazard rate after breast cancer diagnosis), multiple middle variables can exist in the pathways between them [21]. These middle variables, typically called mediators or confounders, help explain the association between the predictor and the outcome [22]. We use multiple mediation analysis to decompose the association to find the indirect effect through each of the middle variables. The remaining effect between the predictor and outcome, not explained by the middle variables, is called direct effect. Figure 2 shows the conceptual model to explore the racial disparity in breast cancer survival. In this paper, we call all middle variables used to explain the racial disparity as the exploratory variables.

We use the multiple mediation analysis method by Yu et al. to differentiate the effects. The method depends on three assumptions as described in [10,23].

The average total effect (TE) is defined as the average change rate of Y with respect to X [10,24]. The average direct effect (DE) not from the ith exploratory variable (M_i_) is calculated as the TE but fixing M_i_ at its marginal distribution when X changes. The average indirect effect (IE) for M_i_ is the difference between the average TE and the average DE not from M_i_. Readers are referred to [10,24] for details of the multiple mediation analysis method.

The outcome variable is time-to-event, where the event is death, and the outcome is right-censored. Both linear (Cox Proportional Hazard) and non-linear (multivariate additive regression trees (MART) models were fit to explore risk factors related to the survival rate [21,24].

The R package “mma” (version 10.7-1) was used for the multiple mediation analysis [25]. The bootstrap method estimates the variances of estimates. A total of 2000 bootstrap samples were used, and the quantiles of bootstrap estimates were used to obtain the confidence intervals.

## 3. Results

The analysis was performed in two steps. First, we identified socioeconomic and demographic variables that significantly explain the observed racial disparity in breast cancer survival. Then, we identified genes that act as exploratory variables to explain the racial disparity among breast cancer participants, adjusting for the risk factors found in Step 1.

### 3.1. Racial Disparity in Breast Cancer Survival

Survival time was measured in days. Participants who had not died were considered as right-censored. The censoring time is calculated as the difference between the earliest condition date and the latest condition date. If there is only one condition date, we imputed the censoring days as five, assuming each participant lived at least five days after breast cancer diagnosis [26].

We performed the log-rank test to check the overall disparity in survival between Black and White participants with breast cancer. Table 2 shows there was a significant difference in the survival time (*p*-value = 0.045), with White participants living on average longer periods after diagnosis. Appendix A shows the Kaplan–Meier plot for overall survival among breast cancer participants by race.

### 3.2. Socioeconomic Determinants Analysis

Based on the literature review and availability of variables in the dataset, the variables listed in Table 1 were considered as potential exploratory variables and covariates. We used two criteria to screen for potential exploratory variables. Firstly, the variable must be significantly correlated with the predictor. Since the predictor is binary (Black vs. White), the criterion is tested using the Chi-square statistics or analysis of variance (ANOVA) method. Secondly, to be identified as a potential exploratory variable, a variable must be significantly related to the outcome after adjusting for all the other variables. The type-III tests with the Cox proportional hazard model are used to check this condition. If the second criterion is not satisfied, the variable is removed for further analysis. If only the second criterion is fulfilled, the variable is considered as a covariate but not as a potential exploratory variable. Table 3 and Table 4 show the descriptive statistics and the *p*-values from both tests. The significance level is set at 0.05 for the exploratory-variable screening.

BMI, age, employment status, general health, and general quality of life were screened as potential exploratory variables. Other variables, including current homeowner, marital status, cigarette 100 lifetime, drug use, highest grade, general physical health, general mental health, social satisfaction, living years at current address, and annual income, were not significantly associated with the outcome. Consequently, they were excluded from further analysis.

### 3.3. Mediation Analysis on Socioeconomic Exploratory Variables

The mediation analysis shows that the variables in Figure 3 partially explain the racial disparity in breast cancer survival rates. The estimated effects with variances and confidence intervals are provided in Appendix A. The nonlinear method in the “mma” R package was used for the estimation.

Based on the results, Black participants have a higher average hazard rate (TE = 0.139, 95% CI (0.038, 0.459) than White participants. The hazard rate is 14.91% (e^0.139^−1) times higher for Black than White participants. The variables age, general health, and general quality have significant indirect effects (*p*-values < 0.001) in explaining the racial disparity. These variables were included in further analysis with genetic factors.

### 3.4. Genomic Data Analysis

As in Section 3.3, genomic data were screened to ensure that only genes with strong signals were selected. First, the univariate Cox proportional hazard model was fit for each gene to screen for gene mutations with significant effects on survival. Genes with *p*-values larger than 0.01 were removed for further consideration. Subsequently, 93 genes were selected. The number of genes for each chromosome is listed in Appendix A.

Next, we performed a multiple mediation analysis on the screened genes for each chromosome. In the analysis, significant socioeconomic exploratory variables—age, general health, and general quality of life—were included. Genes were identified for each chromosome that may explain the observed racial disparity. Appendix A lists the 19 selected genes from each chromosome along with their estimated indirect effects. In the analysis of the 23 chromosomes, the only significant socioeconomic factor was general health.

Lastly, we conducted a multiple mediation analysis considering the 19 genes and the significant socioeconomic risk factor, “general health”, to find the potential exploratory variables for the racial disparity. Table 5 shows the estimated direct and indirect effects.

We then performed the mediation analysis using the non-linear model MART to account for nonlinear associations and potential low-level interactions. Through the analysis, the total effect is significant (TE = 0.593, *p*-value = 0.002, 95% CI (0.187, 0.978)). The direct effect is 0.088 with 95% confidence interval (−0.187, 0.545) containing 0. This means the racial disparity was fully explained by all included exploratory variables. Genes TMEM132B, NARFL, SALL1, PAD12, RIPK1, ASB14, DCX, GNB1L, ARHGAP32, AL135787.1, WBP11, SLC16A12AS1, AP000345.1, IKBKB, and SUPT20H had significant indirect effects at the 5% significance level. “General health” is also significant in this analysis (IE = 0.347, 95% CI (0.206, 0.471)).

To interpret how each variable explains the racial disparity, we can use the visual tools provided in the “mma” R package. Using the gene NARFL as an example, Figure 4 (Left) shows the distribution of the number of mutations in NARFL by race. Compared with White participants, Black participants had a lower average count of mutations. The right panel of Figure 4 shows that the hazard of dying decreased with the number of mutations in NARFL. Therefore, NARFL explains in part the observed racial disparity in breast cancer survival (IE = 0.101, 95% CI (0.032, 0.177)).

Appendix A contain similar graphs for each gene. Genes with positive IEs (in the same direction as the TE) help explain the racial disparity, while genes with negative IEs are protective factors for Black patients. The differences in the number of mutations found in the genes TMEM132B, NARFL, SALL1, PAD12, RIPK1, ASB14, DCX, GNB1L, ARHGAP32, and 263AL135787.1 between Black and White participants, along with how these mutations affect survival rates, help explain the racial disparities observed.

Conversely, the different mutation frequencies in the genes WBP11, SLC16A12AS1, AP000345.1, IKBKB, and SUPT20H between Blacks and Whites along with their mutation effects are linked to survival rates beneficial to Black participants compared to Whites. The disparity would enlarge rather than be explained if those gene mutations were equally distributed between Blacks and Whites.

## 4. Discussion

Despite the comparable incidence rates of breast cancer among Black and White women living in the US, a 42% higher mortality rate is observed among Black breast cancer patients. Our comprehensive analysis using the AoU database underscores the complex interplay of social, environmental, and biological factors that contribute to the higher mortality rate observed among Blacks. In this study, we identified significant differences in the mutation status of 15 genes between Black and White breast cancer participants. Those genes were also linked to breast cancer progression, therefore explaining the observed racial disparity in breast cancer survival.

Interestingly, mutations in five of the identified genes appear to act as protective factors, correlating with improved survival outcomes (hazard rate decreases with the number of mutations). However, only one gene, WBP11, exhibits a higher mutation frequency in Black breast cancer participants compared to White women. This, coupled with the higher mutation rate of the other four protective factors in White women, partially explains the poorer outcomes observed in Black participants.

In comparison, ten identified genes are associated with a worse progression that negatively impacts survival. Notably, two of these genes, TMEM132B and GNB1L, showed a significantly higher mutation status in Black participants. Both TMEM132B and GNB1L have been reported to take part in breast cancer development and progression [27]. Studies have shown that TMEM132B expression can be altered in tumor tissues compared to adjacent healthy tissues. In breast cancer, TMEM132B is associated with poor prognosis and worse clinical outcomes. It implies tumor progression and metastasis [28,29]. Similarly, GNB1L, a member of the G protein family, may impact signaling pathways related to cancer cell growth, survival, and metastasis [30,31]. Our analysis suggests that dysregulated activation of TMEM132B or GNB1L may contribute to worse progression, which was particularly prominent in Black breast cancer participants. In-depth research is required to explore the roles of TMEM132B and GNB1L in breast cancer progression, especially their involvement in pathways related to tumor invasion and metastasis. Understanding the differential signaling status of these genes between Black and White breast cancer patients could provide valuable insights into the mechanisms underlying tumor metastasis and facilitate the development of more effective, personalized treatment strategies, particularly in the context of addressing racial disparities in breast cancer outcomes.

Based on the AoU study, we found significant differences in the survival rates between Black and White breast cancer participants. Black women face a worse chance of survival with breast cancer. Through multiple mediation analyses, we identified 15 significant genes that act as exploratory variables in explaining the racial disparity. The genes TMEM132B, NARFL, SALL1, PAD12, RIPK1, ASB14, DCX, GNB1L, ARHGAP32, and AL135787.1 help explain the observed disparity while WBP11, SLC16A12AS1, AP000345.1, IKBKB, and SUPT20H genes were related to better survival rate among Black participants. Identifying the genes helps researchers design experiments to confirm their effects, consequently leading to novel and effective treatments focusing on minority populations.

There are limitations in the study. Although the AoU dataset boasts diversity, how the AoU samples represent the US population is yet to be tested. In addition, the unavailability of complete zip code data pre-2023 limits the ability to link the database with environmental factors. The lack of comprehensive risk factors, information regarding breast cancer stages, and environmental data hinders a thorough analysis. Future endeavors will focus on incorporating environmental risk factors to enrich our analysis.

## 5. Conclusions

This study uses the comprehensive AoU data to identify risk factors that can be used to explain the observed racial disparity in survival rates among breast cancer patients. A total of 15 genes were identified as explanatory variables after adjusting for socioeconomic factors. Understanding the genetic and nongenetic risk factors of racial differences allows for the development of more personalized treatments and interventions to reduce racial disparities. Specific gene mutations are more prevalent in certain racial groups. For example, TMEM132B and GNB1L are more prevalent in Black participants. Targeted therapies can be developed to address these mutations directly. This research can also lead to broader discoveries that benefit all racial groups by improving our general understanding of breast cancer biology. In future research, we plan to validate the gene effects through in vivo and in vitro experiments, along with finding the pathways through which these genes influence breast cancer outcomes. Moreover, subtyping breast cancer may contribute significantly to our understanding of racial disparity, we intend to design experiments and leverage cancer registry data for stratified analyses of cancer subtypes.

## Figures and Tables

**Figure 1 cancers-16-03294-f001:**
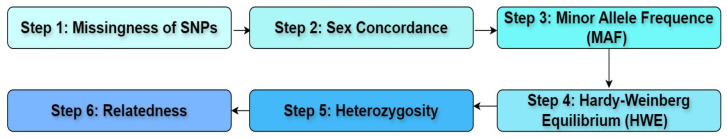
The quality control steps performed on the genomic data.

**Figure 2 cancers-16-03294-f002:**
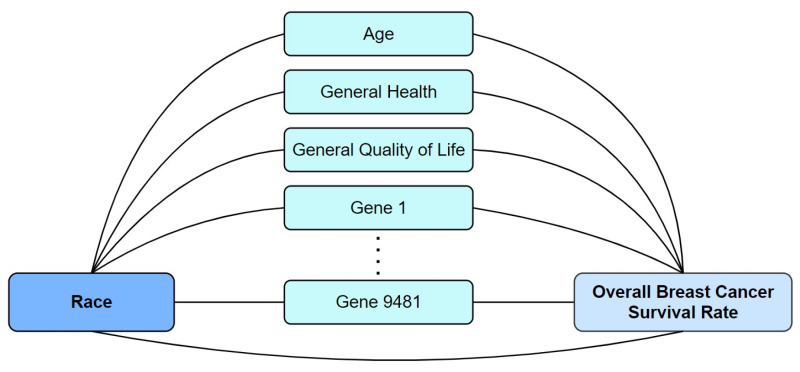
The conceptual model exploring the relationship between race and survival rate among participants with breast cancer through different exploratory variables.

**Figure 3 cancers-16-03294-f003:**
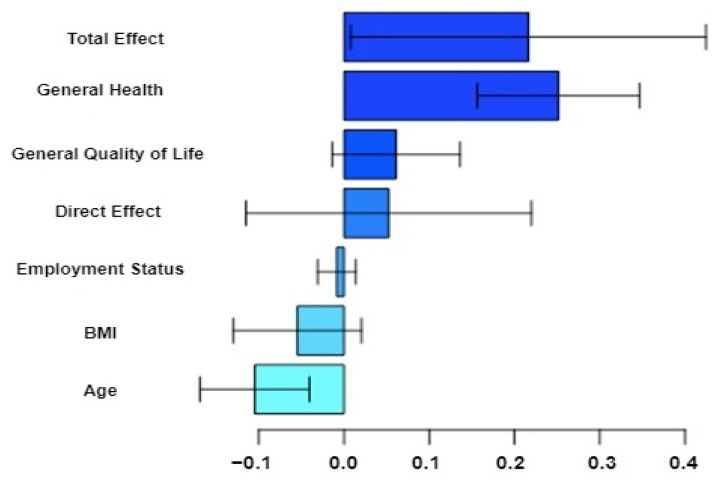
Estimated exploratory variable effects with 95% confidence intervals on racial disparity in breast cancer survival.

**Figure 4 cancers-16-03294-f004:**
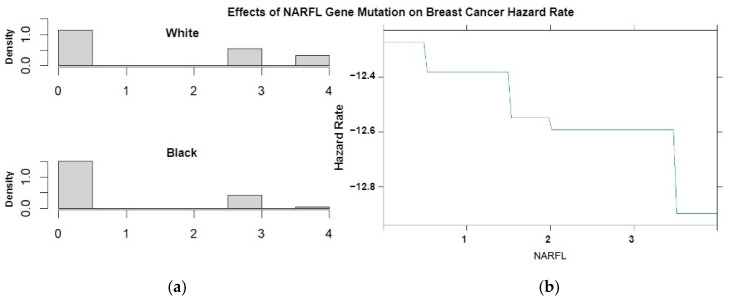
(**a**) The density of the number of mutations for the gene NARFL by race. (**b**) The breast cancer hazard rate by the number of mutations of gene NARFL from Chromosome 16.

**Table 1 cancers-16-03294-t001:** List of variables, descriptions, formats, and sources.

Variable Name	Description	Value/Format	Data Domain
Outcome:			
Breast Cancer Survival	Number of days from diagnosis to either death (event) or latest contact (censor). If only one date is reported, impute 5 days.	Continuous	Condition Data
Vital Status	Indicator of death or not	1-Yes 0-Alive	Death Data (Controlled tier)
Exposure Variable:			
Race	Race of participant	White Black	Person Data
Exploratory Variables:			
Age	Age at diagnosis	Continuous	Calculated
BMI	Body mass index	Continuous	Physical Measurement
Annual Income	Annual household income	35k to 100k Less than 35k More than 100k	Survey Data
Current Homeowner	Owns the home they live in	No Yes	Survey Data
Years Lived at Current Address	Years lived at current address	Less than 5 More than 5	Survey Data
Marital Status	Current marital status	Married/Partner Single (never married, widowed, divorced, separated)	Survey Data
Smoking	Smoked at least 100 cigarettes in entire life	No Yes	Survey Data
Drug Use	Substances used	No Yes	Survey Data
Highest Grade	Highest grade completed	College or above High school (9–12) School 1 to 8	Survey Data
Employment Status	Current employment status	Employed Unemployed	Survey Data
General Health	Health conditions in general	Excellent/Very Good Good Poor/Fair	Survey Data
General Physical Health	Physical health in general	Excellent/Very Good Good Poor/Fair	Survey Data
General Mental Health	Mental health including mood and thinking ability	Good and above Poor/Fair	Survey Data
General Quality of Life	Quality of life in general	Excellent/Very Good Good Poor/Fair	Survey Data
Social Satisfaction	Social satisfaction with social activities and relationships	Excellent/Very Good Good Poor/Fair	Survey Data

**Table 2 cancers-16-03294-t002:** Summary statistics and Log-rank test for survival by race.

Race	Mean (Days)	Median (Days)	*p*-Value
White Participants	1887	1384	0.045
Black Participants	1663	1176	

**Table 3 cancers-16-03294-t003:** Potential Categorical Socioeconomic Exploratory Variables and Covariates.

Variables (n = 7452)	Black (%)	White (%)	*p*-Value 1	*p*-Value 2
Annual Income			<0.001	0.35
35k to 100k	31.80%	40.50%		
Less than 35k	58.90%	19.10%		
More than 100k	9.35%	40.40%		
Current Homeowner			<0.001	0.96
No	59.50%	20.20%		
Yes	40.50%	79.80%		
Years Lived in Current Address			<0.001	0.06
Less than 5	38.60%	29.00%		
More than 5	61.40%	71.00%		
Marital Status			<0.001	0.43
Married or with partner	26.70%	59.50%		
Single	73.30%	40.50%		
Smoking			0.85	0.13
No	58.50%	58.10%		
Yes	41.50%	41.90%		
Drug Use			0.001	0.54
No	49.40%	44.10%		
Yes	50.60%	55.90%		
Highest Grade			<0.001	0.63
College or above	67.40%	88.60%		
High School	31.30%	11.20%		
School 1 to 8	1.24%	0.18%		
Employment Status *			<0.001	0.04
Employed	29.60%	38.90%		
Unemployed	70.40%	61.10%		
General Health *			<0.001	0.01
Excellent or very good	24.00%	47.20%		
Good	37.50%	34.90%		
Poor or fair	38.50%	17.80%		
General Physical Health			<0.001	0.07
Excellent or very good	21.50%	43.70%		
Good	40.20%	36.40%		
Poor or fair	38.30%	19.90%		
General Mental Health			<0.001	0.08
Good and above	81.90%	91.80%		
Poor	18.10%	8.16%		
General Quality of Life *			<0.001	0.01
Excellent or very good	39.20%	69.00%		
Good	41.30%	23.20%		
Poor or fair	19.50%	7.84%		
Social Satisfaction			<0.001	0.17
Excellent or very good	45.50%	65.10%		
Good	32.50%	24.10%		
Poor or fair	22.00%	10.80%		

* screened as potential exploratory variables. *p*-Value 1: Chi-square test of association between race and row variables. *p*-Value 2: Type-3 tests from the Cox-PH model adjusting for all variables.

**Table 4 cancers-16-03294-t004:** Potential Continuous Socioeconomic Exploratory Variables and Covariates.

Variables (n = 7452)	Black Mean (SD)	White Mean (SD)	*p*-Value 1	*p*-Value 2
Age *	57.30 (11.10)	59.60 (11.70)	<0.001	0.01
BMI *	32.70 (7.66)	28.70 (6.85)	<0.001	<0.001

* indicates potential exploratory variables. *p*-Value 1: ANOVA test of row variables with race. *p*-Value 2: Type-3 tests from the Cox-PH model adjusting for all variables.

**Table 5 cancers-16-03294-t005:** Summary of estimated exploratory variable effects for racial disparity in breast cancer survival.

Variable Name	Indirect Effect	SD	95% CI	*p*-Value
TMEM132B *	0.154	0.050	(0.077, 0.275)	<0.001
NARFL *	0.101	0.038	(0.032, 0.177)	0.001
SALL1 *	0.069	0.040	(0.010, 0.167)	0.02
PADI2 *	0.071	0.040	(0.006, 0.158)	0.02
RIPK1 *	0.072	0.031	(0.003, 0.130)	0.01
ASB14 *	0.053	0.038	(0.001, 0.142)	0.04
DCX *	0.057	0.024	(0.015, 0.109)	0.001
GNB1L *	0.038	0.031	(0.000, 0.122)	0.048
ARHGAP32 *	0.028	0.021	(0.000, 0.082)	0.05
AL135787.1 *	0.022	0.015	(0.002, 0.059)	0.02
ADCY1	−0.046	0.030	(−0.115, 0.007)	0.08
WBP11 *	−0.051	0.034	(−0.145, −0.016)	0.002
SLC16A12AS1 *	−0.074	0.036	(−0.170, −0.028)	<0.001
AP000345.1 *	−0.075	0.036	(−0.171, −0.027)	0.002
IKBKB *	−0.111	0.032	(−0.171, −0.045)	<0.001
SUPT20H *	−0.211	0.089	(−0.428, −0.083)	<0.001
General Health *	0.347	0.069	(0.206, 0.471)	<0.001
Direct Effect	0.088	0.184	(−0.187, 0.545)	0.73
Total Effect	0.593	0.199	(0.187, 0.978)	0.005

(*) indicate variables with significant indirect effects at the 5% significance level.

## Data Availability

The data underlying this article are available in the All of US database at https://www.researchallofus.org, and can be accessed by registering for a controlled tier.

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
