# Peer review of "Genomic and Socioeconomic Determinants of Racial Disparities in Breast Cancer Survival: Insights from the All of Us Program"

_cancers, 2024, doi:10.3390/cancers16193294_

Round 1

Reviewer 1 Report

Comments and Suggestions for Authors

1. Figure 1 is small and illegible, it should be redone.

2. In Figure 3, what does "total" and "de" mean? Is this the total effect and direct effect? ​​Add an explanation in the caption under the figure.

3. In Figure 4, the size of the inscriptions should also be increased, since the figure is illegible in its current form.

4. Maybe Figures 1 and 2 should also be made in color, similar to Figure 3? I think it will be more effective.

5. Why didn't the authors conduct a multivariate survival analysis taking into account both socioeconomic and genomic variables to show how comparable the contribution of each is to survival rates? And in general, the Kaplan-Meier curves for comparing survival rates by race are not provided, I would like to see and compare the median survival rates. In the provided analysis, it is unclear whether the predictors are independent or not.

Author Response

1. Summary

Thank you so much for taking the time to review this manuscript. Please find the detailed responses below and the corresponding revisions highlighted in the re-submitted files.

2. Questions for General Evaluation

Reviewer’s Evaluation

Response and Revisions

Are the results clearly presented?

Can be improved

Slight changes were made to make it clearer.

Are the conclusions supported by the results?

Can be improved

3. Point-by-point response to Comments and Suggestions for Authors

Comments 1:  Figure 1 is small and illegible, it should be redone.

Response 1: We agree. Figure 1 (line 116 -117) is redone in the updated manuscript.

Comments 2: In Figure 3, what does "total" and "de" mean? Is this the total effect and direct effect? ​​Add an explanation in the caption under the figure.

Response 2: Thank you for pointing this out. Here, "total” means total effect and “de” is direct effect. Figure 2 (line 215 -216) is redone in the updated manuscript with clearer labels.

Comments 3: In Figure 4, the size of the inscriptions should also be increased, since the figure is illegible in its current form.

Response 3: Agreed. We modified Figure 4 (line 257 -259) so that the inscriptions are clearer.

Comments 4: Maybe Figures 1 and 2 should also be made in color, similar to Figure 3? I think it will be more effective.

Response 4: Thank you for the suggestion. We changed Figure 1 and Figure 2 to have similar color scheme.

Comments 5: Why didn't the authors conduct a multivariate survival analysis taking into account both socioeconomic and genomic variables to show how comparable the contribution of each is to survival rates? And in general, the Kaplan-Meier curves for comparing survival rates by race are not provided, I would like to see and compare the median survival rates. In the provided analysis, it is unclear whether the predictors are independent or not.

Response 5: Thank you for highlighting this. We performed multivariate survival analysis adjusting for all covariates and mediators separately and for the mediation analysis. P-values 2 in Table 3 are the p-values from the Cox-PH model. We added the Kaplan-Meier curves in Supplement Figure S1. We also added Table 2 (line 181) with mean, median survival days for comparison. The predictors are considered as independent in the analysis.

Reviewer 2 Report

Comments and Suggestions for Authors

       Authors in this manuscript studied the Genomic and Socioeconomic Determinants of Racial Disparities in Breast Cancer Survival. Authors also discussed the limitations of their study. I have the following comments which should be addressed before the manuscript can be accepted for publication.

1.        There is uneven spacing between the words in many sentences. Authors should correct this formatting issue.

2.   Line 261-264- “Overall, the different distributions of mutation numbers between Black and White participants within genes TMEM132B, NARFL, SALL1, PAD12, RIPK1, ASB14, DCX, GNB1L, ARHGAP32, and 263AL135787.1 together with the genes’ mutation effects on survival rates explained the observed racial disparity”. This sentence should be rephrased so that it is clearer to understand.

3.   The authors should label the y axis in figure 4B and for the figures for other genes in the supplementary section.

TRANSLATE with x English
Arabic Hebrew Polish
Bulgarian Hindi Portuguese
Catalan Hmong Daw Romanian
Chinese Simplified Hungarian Russian
Chinese Traditional Indonesian Slovak
Czech Italian Slovenian
Danish Japanese Spanish
Dutch Klingon Swedish
English Korean Thai
Estonian Latvian Turkish
Finnish Lithuanian Ukrainian
French Malay Urdu
German Maltese Vietnamese
Greek Norwegian Welsh
Haitian Creole Persian  
TRANSLATE with COPY THE URL BELOW Back EMBED THE SNIPPET BELOW IN YOUR SITE Enable collaborative features and customize widget: Bing Webmaster Portal Back

Author Response

1. Summary

Thank you so much for taking the time to review this manuscript. Please find the detailed responses below and the corresponding revisions highlighted in the re-submitted files.

2. Point-by-point response to Comments and Suggestions for Authors

Comments 1:  There is uneven spacing between the words in many sentences. Authors should correct this formatting issue.

Response 1: Thank you for highlighting it. The uneven spacing is due to the justified text format of the paragraphs as suggested in the manuscript template. Any extra spacing found was removed.  

Comments 2: Line 261-264- “Overall, the different distributions of mutation numbers between Black and White participants within genes TMEM132B, NARFL, SALL1, PAD12, RIPK1, ASB14, DCX, GNB1L, ARHGAP32, and 263AL135787.1 together with the genes’ mutation effects on survival rates explained the observed racial disparity”. This sentence should be rephrased so that it is clearer to understand.

Response 2: We agree. The sentence (line 263-267) was rewritten to make it more understandable as follows:

“The differences in the number of mutations found in the genes TMEM132B, NARFL, SALL1, PAD12, RIPK1, ASB14, DCX, GNB1L, ARHGAP32, and 263AL135787.1 between Black and White participants, along with how these mutations affect survival rates, help explain the racial disparities observed.”

Comments 3: The authors should label the y axis in figure 4B and for the figures for other genes in the supplementary section.

Response 3: Thank you for pointing this out. We modified Figure 4 (line 257 -259) and supplement figures S2-15 with y axis labelled as “Hazard Rate”.

Round 2

Reviewer 1 Report

Comments and Suggestions for Authors

I have no more comments on the manuscript.